# Investigation of Exposure to Occupational Noise among Forestry Machine Operators: A Case Study in Brazil

Diego Aparecido Camargo [1], Rafaele Almeida Munis [1] and Danilo Simões [2,*]

1   School of Agriculture, São Paulo State University (Unesp), Botucatu 18610-034, Brazil;
    diegocamargoflorestal@gmail.com (D.A.C.); rafaele.munis@gmail.com (R.A.M.)
2   Campus of Itapeva, São Paulo State University (Unesp), Itapeva 18409-010, Brazil
*   Correspondence: danilo.simoes@unesp.br

**Abstract:** In mechanized harvesting of wood operations, in a cut-to-length system, occupational noise is emitted by self-propelled forest machines, which compromises the safety and health of operators. Therefore, the occupational noise levels emitted by self-propelled forestry machines, in a cut-to-length system, were investigated to determine which are in line with current Brazilian legislation. The noise levels issued by 22 self-propelled forestry machines in the mechanized harvesting of wood operations, in *Eucalyptus* and *Pinus* planted forests, were collected during a full day of measurement. Taking into account the operations performed and the type of planted forest, homogeneous groups of operators were formed. Based on Regulatory Norms N.9 and N.15 adopted for labor purposes in Brazil, occupational noise levels were analyzed. In relation to harvester operators, 36.4% were exposed to values above the exposure limit of 85 dB (A) and 63.6% to the action level of 80 dB (A). Regarding the forwarder operators, 100% were exposed to values that exceeded the action level. For the analyzed conditions, for the cut-to-length system employed in harvesting wood in forest planted with *Eucalyptus* and *Pinus*, the occupational noise levels of the self-propelled forest machines are not in line with current Brazilian legislation for labor purposes.

**Keywords:** worker's health; ergonomic risk; hearing protection; cut-to-length logging; *Eucalyptus*; *Pinus*; forest management





## 1. Introduction

The mechanization of operations involving the harvesting of wood has become essential for the constant supply of raw materials to the flower-based industries. In Brazil, among the commonly used systems for mechanized harvesting of forests planted with the genera *Eucalyptus* and *Pinus*, the cut-to-length system stands out. This is comprised of self-propelled forestry machines that jointly carry out cutting-sequence operations, processing felled trees and transporting shortwood suspended in bunks.

This system is usually composed of two self-propelled forestry machines, the harvester and forwarder [1–3]. Harvesters perform tree cutting operations and process the felled trees into logs [4–6]. Forwarders carry the shortwood, suspended in bunks, to the margins of forest roads or to intermediate yards [7,8]. As it is a fully mechanized system, it allows for productivity adjustments due to changes in industrial demand, with subsequent productivity gains in forestry operations [9–12].

However, the mechanized harvesting of wood carried out by cut-to-length processing can present ergonomic risks, causing the emergence of occupational diseases. Therefore, it is considered an unhealthy environment that exposes operators to physical and psychological disorders [13–16]. Among the physical agents, occupational noise [17,18] stands out, characterized as sound pressure perceived by the human ear as an undesirable sound with different frequencies, intensities and phases [19–21].

The combination of prolonged exposure to noise levels above what is allowed is one of the factors that aggravates hearing loss, permanently or temporarily [22–24]. In this

perspective, the time of noise exposure at work, associated with genetic factors, leads to the manifestation of complex disorders such as occupational noise-induced hearing loss [25–28].

These disorders cause not only hearing loss, but also memory impairment among other occupational diseases [29,30], e.g., cardiovascular diseases, traumatic injuries, depression, interference in communication and concentration, in addition to physical and physiological stress [31–35].

The identification of the noise-emission levels from each self-propelled forestry machine [36] allows forest managers to implement mitigating actions to protect operators' hearing loss. Our hypothesis is that the occupational noise levels emitted by self-propelled forestry machines are higher than the limits indicated in the Brazilian legislation.

Thus, we investigated if the occupational noise levels emitted by self-propelled forestry machines, in a cut-to-length system, are in line with the current Brazilian legislation for labor purposes.

## 2. Materials and Methods

### 2.1. Study Characterization

As this was a study involving human beings, the research and the respective informed-consent forms were previously approved by the Research Ethics Committee of São Paulo State University (Unesp), Medical School, Botucatu, registered under the number of Opinion 3,492,969.

The data were collected between August and September of 2019 during the mechanized harvesting of wood in Brazilian forests of the genus *Eucalyptus* and the genus *Pinus*. The mean meteorological conditions in the study region, according to the Instituto Nacional de Meteorologia (INMET) [37], were an air humidity of 69.24%, a wind speed of 4.29 ms$^{-1}$ and an air temperature of 289.3 K.

The forest planted with *Eucalyptus* had a mean individual tree volume of $0.29 \pm 0.12$ m$^3$, spacing at 3.3 m $\times$ 1.8 m, and $8.8 \pm 2.3$ years of age. The forest planted with *Pinus* had a mean individual tree volume of $0.64 \pm 0.04$ m$^3$, spacing at 3.3 m $\times$ 1.8 m, and $19.3 \pm 0.22$ years of age. In addition, the relief classes were classified according to Viel et al. [38], as shown in Table 1.

**Table 1.** Study area characterization.

| Genus | Study Area | Size Cutting Area (ha) | Diametric Distribution (cm) | Mean Individual Tree Volume (m$^3$) | Cutting Age (Year) | Terrain Slope (%) | Relief Classification |
|---|---|---|---|---|---|---|---|
| *Eucalyptus* | 1 | 4.22 | 11–11.9 | 0.43 | 11.4 | 10 to 15 | Wavy |
| | 2 | 6.16 | 11–11.9 | 0.32 | 11.0 | 10 to 15 | Wavy |
| | 3 | 1.22 | 10–10.9 | 0.39 | 10.9 | 15 to 20 | Wavy |
| | 4 | 2.88 | 6–6.9 | 0.12 | 6.5 | 20 to 25 | Strong wavy |
| | 5 | 1.13 | 6–6.9 | 0.12 | 6.5 | 27 to 30 | Strong wavy |
| | 6 | 0.84 | 6–6.9 | 0.35 | 6.3 | 30 to 40 | Strong wavy |
| *Pinus* | 7 | 6.23 | 18–19.9 | 0.66 | 19.0 | 27 to 30 | Strong wavy |
| | 8 | 5.30 | 18–19.9 | 0.54 | 19.0 | 33 to 35 | Strong wavy |
| | 9 | 2.27 | 18–19.9 | 0.66 | 19.0 | 27 to 30 | Strong wavy |
| | 10 | 3.54 | 18–19.9 | 0.66 | 19.0 | 20 to 25 | Strong wavy |
| | 11 | 4.61 | 18–19.9 | 0.66 | 19.0 | 5 to 8 | Smooth wavy |
| | 12 | 4.39 | 19–19.9 | 0.65 | 19.0 | 27 to 30 | Strong wavy |

### 2.2. Characteristics of Self-Propelled Forestry Machines and Operators

Dosimetry was performed on 11 harvester operators and 11 forwarder operators. The average age of the harvester operators was $35.4 \pm 7.9$ years, with an average length of experience in the role of $3.6 \pm 1.5$ years. Forwarder operators had an average age of $40.1 \pm 7.0$ years, with an average experience time of $3.5 \pm 5.4$ years.

The harvesters evaluated were Ponsse and John Deere and the forwarders were Ponsse (Table 2). Thus, the wheel systems of all the self-propelled forestry machines were pneumatic with $8 \times 8$ traction.

**Table 2.** Identification, brand and model of self-propelled forestry machines and accumulated hours of use.

| Forest Planted | Self-Propelled Forest Machines | Identification | Brand | Model | Accumulated Hours of Use (h) |
|---|---|---|---|---|---|
| *Pinus* | Harvester | HVP1 | Ponsse | Bear | 5740 |
| | | HVP2 | Ponsse | Bear | 6110 |
| | | HVP3 | Ponsse | Bear | 6200 |
| | Forwarder | FWP1 | Ponsse | Elephantking | 6276 |
| | | FWP2 | Ponsse | Elephantking | 6360 |
| | | FWP3 | Ponsse | Elephantking | 13,352 |
| | | FWP4 | Ponsse | Elephantking | 6380 |
| *Eucalyptus* | Harvester | HV1 | Ponsse | Ergo | 13,547 |
| | | HV2 | Ponsse | Ergo | 11,214 |
| | | HV3 | Ponsse | Ergo | 13,980 |
| | | HV4 | Ponsse | Ergo | 14,600 |
| | | HV5 | Ponsse | Ergo | 14,278 |
| | | HV6 | Ponsse | Ergo | 12,790 |
| | | HV7 | Ponsse | Ergo | 12,366 |
| | | HV8 | John Deere | 1270 | 274 |
| | Forwarder | FW1 | Ponsse | Elephantking | 10,781 |
| | | FW2 | Ponsse | Elephantking | 15,155 |
| | | FW3 | Ponsse | Elephantking | 15,880 |
| | | FW4 | Ponsse | Elephantking | 15,658 |
| | | FW5 | Ponsse | Elephantking | 11,700 |
| | | FW6 | Ponsse | Elephantking | 11,450 |
| | | FW7 | Ponsse | Elephantking | 11,093 |

HVP: harvesters operating in forest planted with *Pinus*; FWP: forwarders operating in forest planted with *Pinus*; HV: harvesters operating in forest planted with *Eucalyptus*; FW: forwarders operating in forest planted with *Eucalyptus*.

### 2.3. Parameterization of Dosimetry

Dosimetry data were collected under normal working conditions, that is, during eight-hour workdays, with one-hour breaks for lunch allowing for tolerances due to physiological needs or mechanical interruptions.

In order to measure the sound pressure level (SPL) received by the operators of the self-propelled forestry machines, two integrating meters for personal use of the Instrutherm brand, models DOS-500 and DOS-600, equipped with regular calibration certificates, were used.

The procedures for collecting SPL followed the guidelines of "Acoustics—Determination of occupational noise exposure Engineering method", the International Organization for Standardization ISO 9612: 2009 [39].

The SPL responses obtained were expressed in decibels (dB), measured every 60 s, adjusted to curve A, as this is the necessary weight to simulate the noise received by the human ear, according to Boger et al. [40]. Receptive responses by the devices were limited to a minimum value of 70 dB (A) and a maximum value of 140 dB (A), given the original configurations of the integrating meters for personal use.

Average levels of daily exposure to occupational noise ($L_{avg}$) were verified, as provided by the guidelines founded by Brazilian legislation for labor purposes, which addresses unhealthy activities and operations under the environmental risk prevention program.

For the eight-hour workday, the criterion of 85 dB (A) was adopted, according to Regulatory Standard N.15—Annex N.1, which establishes the tolerance limits for continuous noise [41]. In addition, Regulatory Standard N. 1733—Appendix N. 6 [42] was used to

estimate the percentage risk of hearing loss for operators with professional experience over five years and with exposure to occupational noise levels over 80 dB (A).

With a view to preventive actions, half the dose was adopted as an action level for the noise physical agent as an action level criterion, supported by Regulatory Standard N.9 [43] and by Directive 2003/10/EC [44], which establishes the lower exposure action value at 80 dB (A). Thus, to obtain the sound pressure level (SPL), Equation (1) was applied according to Kuehn [45].

$$\text{SPL} \; = \; 20\log\left(\frac{\text{P}}{\text{P}_0}\right) \tag{1}$$

where SPL is the sound pressure level, P is the sound pressure being measured and $\text{P}_0$ is the reference sound pressure (standardized at $2 \times 10^{-5}$ Pascal).

Therefore, the average level of daily exposure to occupational noise was estimated, using Equation (2), recommended by Regulatory Standard N.15 [41], for the dose increment factor of 5.

$$L_{\text{avg}} = \; 16.61 \; \times \; \log\left[\frac{\text{D} \; \times \; \text{TC}}{100 \; \times \; \text{TM}}\right] + \text{LC} \tag{2}$$

where $L_{\text{avg}}$ is the average level of daily exposure to occupational noise, D is the daily noise dose projected for 8 h, TC is the base level period of the evaluation criteria (8 h), TM is the measurement time, LC is the base level of the criterion equal to 85 dB (A).

The maximum allowable daily exposure was measured using Equation (3), which was adapted from NIOSH standards for occupational noise exposure [46], considering the action level of Regulatory Standard N.9 [43] and the dose increment factor of 5 [41].

$$\text{MEDP} \; = \; \frac{480}{2^{(L_{\text{avg}} \; - \; 80)/5}} \tag{3}$$

where MEDP is the maximum allowable daily exposure.

Finally, according to Schulz [47], the estimated noise level that the operator's ear captures was measured after attenuation provided through the use of hearing protection devices according to the Noise Reduction Rate Subject Fit (Equation (4)).

$$\text{NP} \; = \; L_{\text{avg}} \; - \; \text{NRRsf} \tag{4}$$

where NP is the estimated noise level that reaches the worker's ear in dB (A), NRRsf is the noise reduction rate subject fit.

*2.4. Determination of Homogeneous Groups*

The homogeneous groups (GH) were determined, allocating the operators of self-propelled forestry machines by the type of operation performed and the type of planted forest in which they were inserted, based on the guidelines of ISO 9612: 2009 [39] which were characterized by

Homogeneous Group 1 (GH1): composed of three self-propelled forestry machines and therefore three operators of harvesters which operate in forests planted in *Pinus*.

Homogeneous Group 2 (GH2): composed of four self-propelled forestry machines and therefore four forwarder operators who operate in forests planted in *Pinus*.

Homogeneous Group 3 (GH3): composed of eight self-propelled forestry machines and therefore eight harvester operators who operate in *Eucalyptus*-planted forests.

Homogeneous Group 4 (GH4): composed of seven self-propelled forestry machines and therefore seven forwarder operators who operate in *Eucalyptus*-planted forests.

The strategic measurement criterion was over a full day, so noise levels were obtained for a daily workday, by homogeneous group, followed by widened uncertainty. This stems from the uncertainties associated with sampling occupational noise levels at work resulting from sampling and instrumentation errors. These noise levels were derived from

the average levels of daily exposure to occupational noise according to ISO 9612: 2009 [39] (Equation (5)).

$$L_{p,A,eqTe} = 10 \log \left( \frac{1}{N} \sum_{n=1}^{N} 10^{0.1 \times L_{p,A,eqT,n}} \right) dB \tag{5}$$

where $L_{p,A,eqTe}$ is the A-weighted equivalent continuous sound pressure level for $T_e$, $L_{p,A,eqT,n}$ is the A-weighted equivalent continuous sound pressure level of sample n, N is the total number of job samples.

The daily noise occupational exposure level was obtained using Equation (6) [39].

$$L_{EX,8h} = L_{p,A,eqTe} + 10 \log \left( \frac{T_e}{T_0} \right) \tag{6}$$

where $L_{EX,8\,h}$ is the A-weighted noise exposure level normalized to a nominal 8 h working day, $T_e$ is the effective duration of the working day, $T_0$ is the reference duration $T_0 = 8$ h.

In addition, Equation (7) was applied to determine the uncertainty of the average levels of daily exposure to occupational noise of the measured values [39].

$$u_1^2 = \sqrt{\frac{1}{N-1} \left[ \sum_{n=1}^{N} \left( L_{p,A,eqT,n} - \overline{L}_{p,A,eqT} \right)^2 \right]} \tag{7}$$

where $u_1$ is the standard uncertainty of the energy average of a number of measurements of A-weighted equivalent continuous sound pressure level, $\overline{L}_{p,A,eqT}$ is the arithmetic average of N job samples of the A-weighted continuous equivalent sound pressure level.

The calculation of the combined standard uncertainty for the level of exposure to noise weighted in the daily working day was calculated using Equation (8) [39].

$$u^2 = (L_{EX,8h}) = c_1^2 u_1^2 + c_2^2 \left( u_2^2 + u_3^2 \right) \tag{8}$$

where u is the combined standard uncertainty, $c_1$ is the sensitivity coefficient associated with job noise level sampling, $c_2$ is the sensitivity coefficient associated with measurement instrumentation, $u_2$ is the standard uncertainty due to the instrumentation, and $u_3$ is the standard uncertainty due to microphone position.

Finally, expanded uncertainty (U) was determined using Equation (9) [39].

$$U = 1.65 \times u \tag{9}$$

### 2.5. Statistical Analysis

In order to validate the assumption of statistical equality in the arrangement of the homogeneous groups, the assumptions of data normality and homogeneity of variances were verified using the Lilliefors [48] and Bartlett [49] tests, respectively.

The comparative analysis of sound pressure levels between the homogeneous groups and within each homogeneous group was developed using Friedman's non-parametric ranks test [50]. The implementation of the tests was facilitated through use of the software R, version 3.5.2 [51] and the results were discussed at a significance level of 0.05.

## 3. Results

### 3.1. Exposure to Occupational Noise

Analyzing the harvester operators, it was found that 36.4% of the total operators of self-propelled forestry machines were exposed to occupational noise levels above the exposure limit recommended by Regulatory Standard N.15 [41]. In addition, the other 63.6% were exposed to occupational noise levels above the action level (Table 3) recommended by Regulatory Standard N.9 [43] and Directive 2003/10/EC [44].

**Table 3.** Average levels of daily exposure to occupational noise, maximum allowable daily exposure and protection level.

| Homogeneous Groups | Identification | $L_{avg}$ [dB(A)] | MEDP [h] | NPdB [(A)] |
|---|---|---|---|---|
| GH1 | HVP1 | 84.3 | 8 h 49 | 67.3 |
| | HVP2 | 84.3 | 8 h 50 | 67.3 |
| | HVP3 | 89.6 | 4 h 13 | 72.6 |
| GH2 | FWP1 | 84.4 | 8 h 41 | 67.4 |
| | FWP2 | 84.0 | 9 h 11 | 67.0 |
| | FWP3 | 84.3 | 8 h 50 | 67.3 |
| | FWP4 | 84.3 | 8 h 50 | 67.3 |
| GH3 | HV1 | 83.9 | 9 h 19 | 66.9 |
| | HV2 | 84.2 | 8 h 56 | 67.2 |
| | HV3 | 84.2 | 8 h 56 | 67.2 |
| | HV4 | 84.4 | 8 h 41 | 67.4 |
| | HV5 | 84.1 | 9 h 03 | 67.1 |
| | HV6 | 86.5 | 6 h 30 | 69.5 |
| | HV7 | 91.2 | 3 h 23 | 74.2 |
| | HV8 | 85.2 | 7 h 46 | 68.2 |
| GH4 | FW1 | 83.9 | 9 h 19 | 66.9 |
| | FW2 | 84.2 | 8 h 56 | 67.2 |
| | FW3 | 84.4 | 8 h 41 | 67.4 |
| | FW4 | 84.2 | 8 h 56 | 67.2 |
| | FW5 | 84.0 | 8 h 19 | 67.0 |
| | FW6 | 84.8 | 8 h 13 | 67.8 |
| | FW7 | 84.1 | 9 h 03 | 67.1 |

HVP: harvesters operating in forest planted with *Pinus*; FWP: forwarders operating in forest planted with *Pinus*; HV: harvesters operating in forest planted with *Eucalyptus*; FW: forwarders operating in forest planted with *Eucalyptus*.

Thus, the HVP3 operator was exposed to an average level of daily exposure to occupational noise at 5.4% above the recommended exposure limit, which caused the reduction of the maximum allowable daily exposure from 100% to 52.7%. However, with the use of hearing protectors, there was an attenuation of 17 dB (A), which reduced the estimated noise level absorbed through the operator's ear to 72.6 dB (A).

The HV6 operator, on the other hand, was 1.7% above the exposure limit; therefore, the maximum allowable daily exposure (MEDP) was 81.2%, which with the attenuation resulting from the use of hearing protectors, resulted in an estimated noise level (NP) of 69. 5 dB (A). The HV7 operator was 7.3% above the exposure limit, with a MEDP of 42.3% and an NP of 74.2 dB (A). Likewise, the HV8 operator was exposed to 0.2% above the exposure limit, with an MEDP of 97.1% and an NP of 68.2 dB (A).

As for the forwarder operators, it was found that none of them were exposed to occupational noise levels higher than the exposure limit, resulting in periods of MEDP above that stipulated for compliance during the daily workday. However, 100% of operators were exposed to noise levels above the action level. Thus, with the use of hearing protectors, the maximum NP was 67.8 dB (A).

In addition, as the HV1 operator had seven years of experience and the FWP4 operator had twenty years, the risk of hearing loss percentages were 1.4% and 5.2%, respectively, according to the Portuguese Standard N.1733 [42].

### 3.2. Exploratory Analysis of the Homogeneous Groups

Based on the normative recommendations of ISO 9612: 2009 [39], the daily noise occupational exposure level in GH1 was 86.8 dB (A), with an expanded uncertainty of $\pm$ 7.1 dB (A). In GH2 the daily noise occupational exposure level was 84.3 dB (A) $\pm$ 3.0 dB (A). In GH3 the level was 86.3 dB (A) $\pm$ 3.5 dB (A) and for GH4, the level was 84.2 dB (A) $\pm$ 3.0 dB (A).

Thus, it was found that there was no statistical difference between the homogeneous groups, for which the assumptions of normality of SPL and the homogeneity of variances were rejected at the level of 0.05 of significance. Therefore, when subjected to Friedman's ranks test, it was found that it was not possible to show a statistical difference between homogeneous groups (*p*-value = 0.9) (Table 4).

**Table 4.** Median and amplitude of the sound pressure level in homogeneous groups.

| Homogeneous Groups | Median [dB(A)] | Amplitude [dB(A)] |
| --- | --- | --- |
| GH1 | 77.9 | 18.8 |
| GH2 | 75.7 | 8.4 |
| GH3 | 77.5 | 9.2 |
| GH4 | 75.9 | 5.4 |

When analyzing the assumptions of normality and homogeneity within the homogeneous groups, they were not accepted at the 0.05 significance level for the self-propelled forestry machines of each homogeneous group. Therefore, through the Friedman ranks test, it was identified that the medians of the SPL of homogeneous groups 1, 2, 3 and 4 did not differ from each other (*p*-values $\geq$ 0.5). Likewise, in GH2 the medians of the SPL did not differ among themselves (*p*-values $\geq$ 0.9), the same persisted for GH3 (*p*-value of 0.9) and therefore, GH4 with *p*-value of 0.9 (Figure 1).

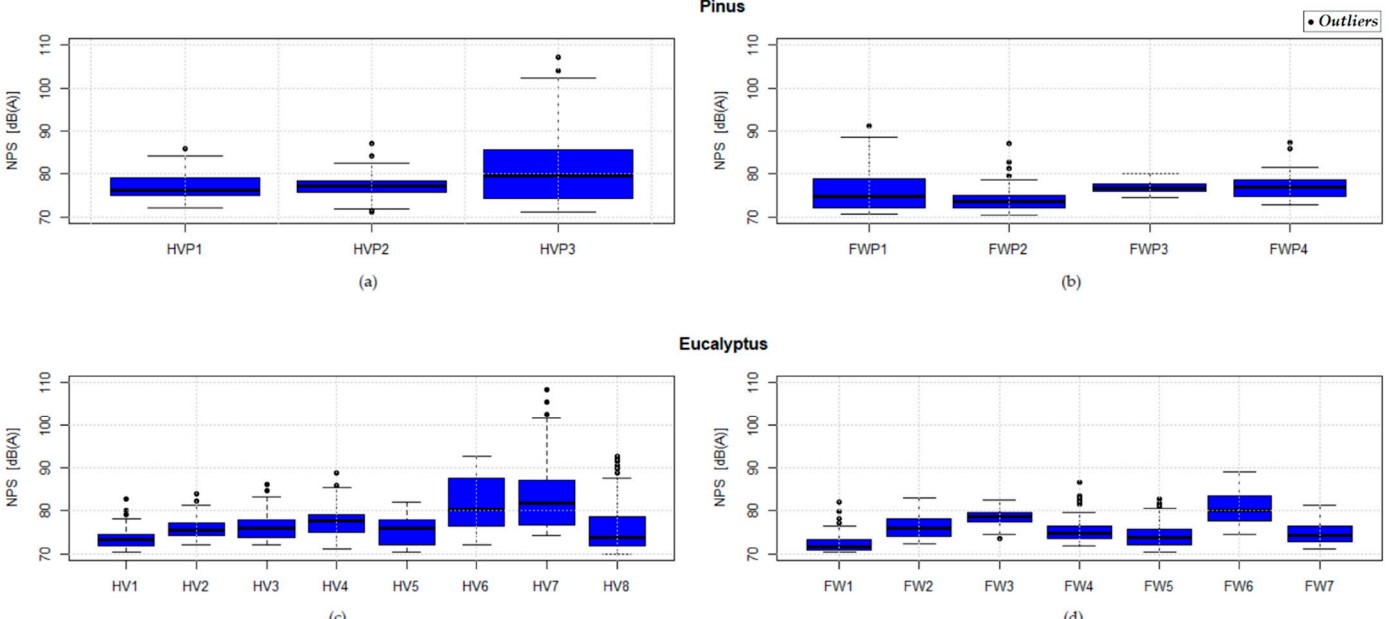

**Figure 1.** Comparative analysis of the elements of each homogeneous group. (**a**) GH1, (**b**) GH2, (**c**) GH3 and (**d**) GH4.

## 4. Discussion

Mechanized harvesting of wood exposes the operators of self-propelled forestry machines to occupational noise levels. In this perspective, Al-arja and Awadallah [52], Fernandes et al. [53] and Fonseca et al. [54] reinforced the need to measure and monitor noise levels as they are considered one of the main causes of damage to occupational health.

According to Nieuwenhuis and Lyons [55] and Kurt et al. [56], by measuring the average levels of daily exposure to occupational noise, it is possible to explain the impact of compliance within operations on the safety and health of operators. Therefore, in the monitoring of the cut-to-length system in *Eucalyptus* and *Pinus* forests, 63.6% of the harvester operators and 100% of the forwarder operators were exposed to values above the 80 dB (A) action level.

Poje et al. [5] observed that the exposure of harvester operators to noise during work ranged from 53.6 to 95.3 dB (A), and the exposure of an operator to noise during forwarding ranged between 50.0 and 108.6 dB (A). Likewise, Sowa and Leszczyński [57] found the exposure of harvester operators to noise during work of 68.4 dB (A) and 73.1 dB (A) for forwarder operators.

As operators were exposed to average levels of daily exposure to occupational noise above the action level, Cheța et al. [58] and Sowa and Leszczyński [57] indicated that operations cannot be performed without the use of hearing protectors, in line with Regulatory Standard N.9 [43] and Directive 2003/10/EC [44]. Thus, the correct use of hearing protectors by insertion and circum-auricular allows for attenuation up to 17 dB and 20 dB, respectively.

In addition, 36.4% of combined operators were exposed to occupational noise levels above the exposure limit. According to Bolaji et al. [18], Rech et al. [59] and Poje et al. [13], in addition to being a critical event, it causes operators to be exposed daily to physical agent noise, with short-term consequences and irreversible effects over the years of exposure.

The noise levels of these self-propelled forestry machines were above the exposure limit determined by Regulatory Standard N.15 [41] of Brazilian legislation for labor purposes. Thus, HVP3, operated in a *Pinus* forest and HV6, HV7 and HV8 operated in a *Eucalyptus* forest, all on land with a slope between 27 and 30% and classified as strong undulating. Rehn et al. [60], Melemez and Tunay [61], Santos et al. [62], and Iftime et al. [35] assessed that on sloping terrain, the engine requirements of self-propelled forestry machines increases, which consequently increases the emitted noise levels.

In addition, problems in sealing the cabin of the HVP3 have been identified in malfunctions caused by falling tree branches during operation. According to Gerasimov and Sokolov [63], Guarnaccia et al. [36], Gerasimov and Sokolov [8], and Lima et al. [64], all self-propelled forestry machines must have sealing systems that allow noise to be confined, promoting a favorable and safe atmosphere for the operator.

The average levels of daily exposure to occupational noise to which these operators were exposed during the eight-hour working day, went beyond the maximum allowable daily exposure threshold. Obtaining the maximum allowable daily exposure level provides a beneficial threshold, depending on the health of the operators, which according to Caruso [65] and Barck-Holst et al. [66], must be attended to with a reduction in the indicated workday hours.

However, when the maximum allowable daily exposure is reached, protection levels can be achieved with the use of hearing protectors. For example, with the use of hearing protectors, the average level of protection for operators was 67.9 dB (A). In addition, the operators of HV1 and FWP4 came with the aggravating factor of having been exposed to occupational noise levels for periods exceeding five years, which may result in hearing loss risks.

Occupational noise-induced hearing loss is characterized by being a silent disease in the short term. However, delay in diagnosis culminates in irreversible effects. As such, the use of hearing protectors to mitigate occupational noise levels that exceed the values stipulated by Portuguese Standard N.1733 [42], acts in a manner that is beneficial to the occupational health of operators. According to Bonnet et al. [67], Borz et al. [22] and Lin et al. [68], the use of hearing protectors favors the attenuation of occupational noise levels, thus fulfilling the requirements of a full eight-hour day.

Measures such as the use of hearing protectors requires the implementation of appropriate decision-making by forest managers. According to Potočnik et al. [69], Noweir and Zytoon [70], Rubio-Romero et al. [71], and McLain et al. [72], these measures should be guided by the monitoring and control of the average levels of daily exposure to occupational noise of all self-propelled forestry machines in forest-harvesting systems.

The allocation of operators in the homogeneous groups allowed us to infer it was not possible to show a statistical difference between the noise levels emitted by self-propelled forestry machines that performed the same or different operations. Therefore, according to

Poje and Potočnik [73], Neitzel et al. [74], Poje et al. [13] and Burella and Moro [75], this strategy allows, in general, for the mitigating measures to be adopted.

Finally, considering that the maximum uncertainty was $\pm 7.1$ dB (A) and that there was no statistical difference between the homogeneous groups, nor in the individual analysis, the daily noise occupational exposure level was above the exposure limit. Poje et al. [5], when obtaining an uncertainty of $\pm 8.4$ dB (A), pointed out that the occupational noise levels close to the exposure limit can disturb operators, affecting productivity.

In this sense, Chapman and Husberg [76], and Akay et al. [77] corroborate that the adoption of standardized measures increases the profitability resulting from the decision-making process and thus allows for the use of generalized actions to other situations. From this perspective, the emission of noise levels from self-propelled forestry machines affect the operator's well-being, health and safety as observed by Bolaji et al. [18] and Iftime et al. [78].

The attention and commitment of forest management when considering workers' labor protection becomes essential and, according to Ghaffariyan et al. [7], Robert et al. [17], and Garmer et al. [79], is mandatory according to the legislation. In this way, agreement between forest management and the criteria defined in the current Brazilian legislation for labor purposes ensures compliance and promotes healthy conditions for operators of self-propelled forestry machines.

Therefore, according to Riccioni et al. [80] and Poje and Mihelič [81], actions such as the elimination, reduction and adequate updating of prevention and protection activities, confer physical benefits to the operators. However, observance of the integrity of the work environment and the monitoring of physical agents, above all occupational noise, is emphasized.

## 5. Conclusions

For the analyzed conditions, the cut-to-length system used for harvesting wood in forests planted with *Eucalyptus* and *Pinus* in Brazil, the occupational noise levels of the self-propelled forestry machines are not in line with current Brazilian legislation for labor purposes.

Mitigating actions should be adopted for the hearing protection of operators of self-propelled forestry machines employed in a cut-to-length system for harvesting wood in *Eucalyptus* and *Pinus* planted forests.

For labor purposes, the adoption of insertion or circum-auricular hearing protectors during the daily workday minimizes the exposure of operators to occupational noise above 80 dB (A), as recommended by current Brazilian legislation.

Periodic training should be carried out with operators to ensure the appropriate use of hearing protectors, in order to reduce the exposure levels of occupational noise.

The individual analysis of the operator of a self-propelled forestry machine employed in harvesting wood in a cut-to-length system can be extrapolated to a group of operators, thereby meeting the requirements of the current Brazilian legislation for labor purposes and providing safety and protection of the occupational health of operators.

**Author Contributions:** Conceptualization, D.A.C. and D.S.; methodology, D.A.C. and D.S.; software, R.A.M.; validation, D.A.C., D.S. and R.A.M.; formal analysis, D.A.C.; investigation, D.A.C. and R.A.M.; resources, D.A.C.; data curation, R.A.M.; writing—original draft preparation, D.A.C.; writing—review and editing, D.A.C. and D.S.; visualization, D.A.C. and D.S. and R.A.M.; supervision, D.S.; project administration, D.S. All authors have read and agreed to the published version of the manuscript.

**Funding:** This research received no external funding.

**Acknowledgments:** This study was financed in part by the Coordenação de Aperfeiçoamento de Pessoal de Nível Superior—Brasil (CAPES)—Finance Code 001.

**Conflicts of Interest:** The authors declare no conflict of interest.

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
