# Peer review of "Investigation of Exposure to Occupational Noise among Forestry Machine Operators: A Case Study in Brazil"

_forests, doi:10.3390/f12030299_

Round 1
Reviewer 1 Report
Dear author(s), please consider following comments/ suggestions:
Line 59 – 61: I understand that was the goal of the research. Please consider to set also research hypothesis. In conclusions the answers should be clearly stated.
Line 63: in chapter 2.1 Study characterization please consider: when the measurements were taken? What was the size of the cutting area? What was the air temperature, air humidity, wind speed, etc. during the measurements? Were conditions similar for all from this point of view?
Line 86: Table 2: please check English
Line 106: … normal working conditions…: please be more specific
Line 120 and Eq. 1: make sure P0 refers to Po in eq. 1 (symbols must be the same)
Line 124: please consider writing Lavg as Lavg (index), similar comment refers to other equations where necessary
Line 205: … according to Portuguese Standard N.1733…: when you basically refer to Brazilian legislation it's difficult to refer to Portuguese Standard and make conclusions on such a basis. On the other hand (in order to higher international soundness of the paper) please consider referring also to e.g. EU standards (Directive 2003/10/EC - Noise) and others
Line 206: Have you recorded e.g. number and total volume of trees harvested - it could be used as a covariate in statistical analysis?
Line 323: … the adoption of insertion or circum-auricular hearing protectors…: Please consider: some machines have already more than 15.000 operating hours. Were they regularly maintained? What about i.e. introducing modern machines with better ergonomic characteristics? Use of ear protection is a short-term solution. On the other hand education of operators in terms of adjusting operating hours and breaks might also be efficient in this scope.
Reviewer 2 Report
General comments
The paper is not much interesting, as it is presented as an employer’s evaluation risk, with a questionable approach (see below). There is in addition, another question: all the described machines are equipped with insulated cabins, but they are useless if the operators work with the windows open (or broken, but this fact could not be possible, as the employer must immediately provide to repair the machine).
Other comments:
Introduction
Line 35-37: useless bibliography (from 1 to 8): the meaning of forwarder and harvester and of their application is widely known.
Line 39: the reference [13] refers to the broiler production, quite different from forestry operations…
Again: references [14] and [15] are not related to the mechanized harvesting. Reference [19] doesn’t deal with occupational noise. Reference 39 doesn’t deal with noise emission levels of any machine.
Line 45: longitudinal frequency??? You may refer to the frequency (number of wavelengths per second) of a longitudinal wave, but a longitudinal frequency is a nonsense.
Lines 52-56: already written in the previous sentences.
Materials and method
Table 1: Eucalyptus and Pinus GENDER???
Table 1 is mainly useless for the paper goal statement.
The Authors title the paragraph 2.2 as ‘Specifications of self-propelled forest machines” and mix with some operators’ characteristics.
Table 2: the title is not in English
Line 117: NPS: what is this?
Equation 1 is wrong. The sound pressure level SPL, A-weighted, is given by the another formula, as described in Mun and Geem, the Authors you cited ([46]). Probably you had in your mind the formula to convert Pascal into dB , but anyway, Equation 1 is not correct.
Line 123: Zytoom is not in the reference chapter.
Equation 2. I’ve not found any reference to Equation 2 in the work you cited [47], and I’ve never seen this equation in other papers.
Equation 3: The Authors of the paper [48] use a different equation. The problem is that the source is unknown.
Equation 4. The measurement of sound attenuation of hearing protectors is not a so easy task. For these reason there are international ISO standard to help the researchers and the technicians: the ISO 4869 series. I suggest the Authors to study these standard to prevent misleading evaluations.
Equations 5, 6, 7, 8 and 9 are reported in the ISO 9612: it is necessary to cite the source.
Line 179: the significance level in the scientific works is written in decimal.
In the material and method chapter it is useful to have a table with the action and limit values of the operators’ daily exposure of the Brazilian regulation, otherwise the results are unintelligible.
In this chapter it is also necessary to provide a detailed measurement chain and methodology used in field.
I don’t understand the mix between the use of not well defined equations (2, 3 and 4) and the daily occupational noise exposure, as defined by the ISO 9612. Perhaps your Brazilian regulation use equations 2, 3 and 4 and not the ISO standard? May you explain it?
Results
The first part could be better be explained in a table. However, I don’t understand why you explained the daily occupational noise exposure, as described in the ISO 9612, and you did not discuss it in the result chapter.
Discussion
The use of the earing protectors to prevent occupational noise induced hearing losses when this risk is present, is known everywhere and a discussion about this aspect is not relevant.
Reference
The reference must be a ‘real’ reference, not a mix of papers concerning different topics.
Reviewer 3 Report
The level of mechanization of forestry work is constantly increasing in the world. Harvesters and forwarders are more and more often used for logging. Machines emit noise and increased exposure of operators to it has a negative effect.
The article refers to noise level determination by CTL harvesting, with harvesters and forwarders, in Eucalyptus and Pinus planted forests. Its layout is typical for scientific articles. Nevertheless, some adjustments should be made to it before publication.
- The Introduction should be somewhat expanded. Although it refers to 39 cited issues, in my opinion, it was written too generally. In principle, it does not present the level of noise emissions for any of the machines, and such research was already carried out in the world. As an example to use, I am giving you one of the European Union-funded projects, "Ergoefficient mechanized logging operations (Ergowood)". You can also use the results published by Sowa and Leszczyński (https://www.formec.org/images/proceedings/2007/session_4_pdf/4_2_paper_sowa_leszczynski_austro_formec_2007.pdf), which also present results of other machines.
The introduction should also end with a clearly defined research goal.
- Results. If possible, please indicate in which activities the noise level was highest (fellings, debranching, or bucking). Was the working day structure also tested? The noise level in relation to individual time fractions would allow a much deeper comparative analysis.
- The discussion was generally written correctly. However, I would like you to suggest introducing additional text and data on comparisons with chainsaws that are still used to a large extent in the world. In addition, the topic of preventive activities, limiting excessive noise, should be presented more broadly.
- In Conclusion, it should be specified to what extent the permissible noise levels are exceeded.
- Author Contributions. Please do not provide full names, only initials.
Round 2
Reviewer 2 Report
Pascal, not Pascals!
Author Response
Dear Reviewer,
I would like to thank you once again for the review and fot the valuable contribution.
Point 1: Pascal, not Pascals!.
Response 1: We agree, and it was fixed (line 131).
Best regards.

Reviewer 3 Report
Most of my comments were not included in the revised version. For this reason, I suggest the authors making further corrections.
